# Hepatitis B Virus Genotype H: Epidemiological, Molecular, and Clinical Characteristics in Mexico

**DOI:** 10.3390/v15112186

**Published:** 2023-10-30

**Authors:** Arturo Panduro, Sonia Roman, Saul Laguna-Meraz, Alexis Jose-Abrego

**Affiliations:** 1Department of Genomic Medicine in Hepatology, Civil Hospital of Guadalajara, “Fray Antonio Alcalde”, Guadalajara 44280, Jalisco, Mexico; sslagna@gmail.com (S.L.-M.); alexisjoseabiology@gmail.com (A.J.-A.); 2Health Sciences Center, University of Guadalajara, Guadalajara 44340, Jalisco, Mexico

**Keywords:** hepatitis B virus, Mexico, prevalence, HBsAg, Anti-HBc, NAT, genotypes, liver cirrhosis, diagnostics, molecular epidemiology

## Abstract

The hepatitis B virus (HBV), comprising of ten genotypes (A-J), has been a silent threat against humanity, constituting a public health problem worldwide. In 2016, the World Health Organization set forth an impressive initiative for the global elimination of viral hepatitis by 2030. As the target date approaches, many nations, particularly in the Latin American region, face challenges in designing and implementing their respective elimination plan. This review aimed to portray the state of knowledge about the epidemiological, molecular, and clinical characteristics of HBV genotype H (HBV/H), endemic to Mexico. PubMed, Scopus, Web of Science, and Google Scholar were searched to compile scientific literature over 50 years (1970–2022). A total of 91 articles were organized into thematic categories, addressing essential aspects such as epidemiological data, risk factors, HBV genotype distribution, HBV mixed infections, clinical characteristics, and vaccination. The prevalence and its associated 95% confidence interval (95% CI) were estimated using the Metafor package in R programming language (version 4.1.2). We provide insights into the strengths and weaknesses in diagnostics and prevention measures that explain the current epidemiological profile of HBV/H. Training, research, and awareness actions are required to control HBV infections in Mexico. These actions should contribute to creating more specific clinical practice guides according to the region’s characteristics. Mexico’s elimination plan for HBV will require teamwork among the government health administration, researchers, physicians, specialists, and civil society advocates to overcome this task jointly.

## 1. Introduction

Hepatitis B virus (HBV) has been a persistent pathogen infecting humans for millennia, causing acute and chronic illnesses [1]. This virus–host interaction is the primary driver of hepatitis, liver cirrhosis, and hepatocellular carcinoma (HCC) pathogenic pathways [2]. Furthermore, it is the mechanism by which HBV evolves and spreads among populations while interacting with environmental factors that need to be understood to prevent and treat this transmissible disease and comprehend the natural history of this infection.

Humanity has transitioned from nomadism to sedentarism, dispersing HBV globally [3,4]. HBV isolates are conventionally classified into nine major genotypes, from A to I, and a tenth putative genotype J, which includes 40 subgenotypes [3,5]. Large-scale paleogenomic investigations of HBV genomes utilizing ancient remains have reconstructed HBV’s phylogeographic history while demonstrating the timing and routes of worldwide human dispersal as far back as the Bronze Age [1,4]. Because of recent migrations, many of these genotypes are identified in the US and Europe. Nonetheless, HBV genotypes are typically associated with a geographic region and host population and, more significantly, have diverse clinical outcomes [6,7,8,9]. Genotype A, for example, predominates throughout Europe and southeastern Africa. Genotypes B and C prevail in most parts of Asia, whereas genotypes I and J (considered a recombination of genotype C) were reported in Vietnam/Laos and Japan, respectively [10,11]; genotype D is found globally but is most frequent in the Mediterranean and Arabic regions, India, and Russia and genotype E in western Africa [5,12]. In the Americas, genotype F is common in Central and South America, whereas genotype H is highly prevalent in Mexico [13]. Finally, genotype G is reported widely among men who have sex with men (MSM) and human immunodeficiency virus (HIV)-infected patients [14], yet it is the second most frequent genotype in Mexico [15]. In conjunction, they inflict 296 million chronic infections, 1.5 million new cases, and 820,000 deaths from liver cirrhosis and HCC each year [16].

This scenario prompted the World Health Organization (WHO) in 2016 to launch the Global Health Strategy against viral hepatitis A-E, mainly focusing on chronic hepatitis B and C, to reduce their incidence to 90% and liver-related mortality to 65% by 2030 [17]. Five strategic directions were established to lead all nations in executing national programs by screening people at risk of infection and identifying subpopulations needing diagnostic testing, treatment, and immunization [18]. Initially, there was much excitement, especially with the new direct-acting antiviral regimens for hepatitis C virus (HCV). However, the path has not been without hurdles, as the Polaris Observatory recently issued a cautious forecast due to the poor progress toward eliminating HBV amongst virtually all 170 nations participating in their modeling study [19]. Initiatives such as the International Coalition to Eliminate HBV (ICE-HBV) are extremely important to advocate for the global cure of HBV by bringing together work groups covering virology, immunology, innovative tools, and clinical trials [20,21].

Latin American countries are under tremendous strain to fulfill the elimination goals because data on HCV prevalence are incomplete, whereas only Mexico, Brazil, Argentina, and Peru have begun national HCV elimination programs [22,23,24]. Mexico launched the “*Plan Nacional para la Eliminación de la Hepatitis C*” in mid-2020, and screening campaigns, micro-elimination, and access to treatment in risk groups are now underway [25,26]. However, HBV elimination strategies are lagging [27]. The absence of a comprehensive surveillance system, a fragmented healthcare system, and a scarcity of research all contribute to this predicament [23]. Furthermore, there is a tendency to transfer data from other regions into local clinical practice guidelines (CPGs) to fill the gaps when it comes to indications for testing, treatment, and prevention measures without considering the host’s and virus’s genetics and environmental factors, which determine the clinical course of infection based on the population’s predominant genotype, which in this case is HBV genotype H [28].

In this review, we present the state of knowledge of HBV infections in Mexico in terms of epidemiology, clinical diagnostics, treatment, and prevention, highlighting the strengths and weaknesses in order to reinforce the actions required to control HBV infections as part of a successful elimination plan. We gathered data from numerous studies published over 50 years (1970–2022) to fully understand the epidemiological, molecular, and clinical characteristics of HBV/H endemic to Mexico, often wrongly considered an HBV genotype indigenous to Central America [13].

## 2. Methods

For this review, we searched the databases PubMed, Scopus, Web of Science, and Google Scholar for the terms “HBV”, “hepatitis B”, “HBV genotypes”, “HBV genotype H”, “epidemiology”, “Mexico”, “clinical characteristics”, “risk factors”, “incidence”, “prevalence”, “mortality”, “morbidity”, “hepatitis B surface antigen, HBsAg”, “antibody against hepatitis B core, Anti-HBc”, “antibody against HBsAg, Anti-HBsAg”, “polymerase chain reaction, PCR”, and “occult hepatitis B, OBI” in English or Spanish. A total of 414 articles were retrieved, of which 198 were excluded because they were case reports, conference abstracts, meeting summaries, personal communications, systematic reviews, or meta-analyses. Subsequently, 125 articles were excluded due to redundancy or a lack of data concerning the prevalence of HBsAg or Anti-HBc. This selection process culminated in including 91 eligible articles, spanning a study period of over 50 years, from 1970 to 2022. These selected articles were subsequently organized into thematic categories, addressing essential aspects such as epidemiological data, risk factors, HBV genotype distribution, HBV mixed infections, clinical characteristics, and vaccination.

The prevalence and its associated 95% confidence interval (95% CI) were estimated using the Metafor package in R programming language (version 4.1.2). This method is commonly used to estimate an overall prevalence while accounting for between-study variability [29].

The incidence of hepatitis HBV was obtained from data provided by the Mexican government, specifically from the official website of the Ministry of Health, https://epidemiologia.salud.gob.mx/anuario/html/incidencia_casos.html (accessed on 5 May 2023). This web portal provided detailed information on the incidence of new cases of diseases, broken down by age and gender, for the period between 2000 and 2021. In order to analyze these data, an average incidence rate was calculated over this period. Subsequently, this information was represented in a bar chart to effectively visualize the evolution of the incidence over the two decades under analysis. Furthermore, data pertaining to new cases of hepatitis B by state for the period spanning from 1995 to 2021 were also gathered from the same source. These data were used to create a state-level gradient map.

Regarding Figure 8, the relationship between HBV viral load levels and advanced liver damage (F3–F4) in patients co-infected with HIV was assessed. We took data comprising 35 HIV/HBV co-infected patients, and liver stiffness measurements alongside HBV viral load data were analyzed [30]. The Receiver Operating Characteristic (ROC) curve analysis established the quantitative value threshold. This analytical procedure was carried out using the Statistical Program for Social Sciences software (SPSS 22.0, IBM, Inc., Armonk, NY, USA).

## 3. An Overview of the Major Milestones of Viral Hepatitis

No narrative would be complete without addressing some research milestones about viral hepatitis and its global relevance for hepatology, allowing healthcare professionals to start looking for the etiological causes of hepatitis in Mexico.

Hippocrates first identified the yellowing of the skin and eyes (jaundice) as a clinical sign of what we know today as viral hepatitis, although it can reflect several other liver diseases [31]. Over time, terms like “campaign jaundice”, “epidemic jaundice”, “infective jaundice”, and “infectious jaundice” were designated. Eventually, these were linked to type A hepatitis [32]. Another term, “catarrhal jaundice”, coined in 1865 by German doctor Karl Virchow, was dismissed after World War II, when “infectious jaundice” or type A was differentiated from “serum hepatitis” or type B hepatitis [33], iatrogenically contracted by administering tainted vaccines and blood transfusions.

The following 60 years of research led to a comprehensive understanding of the immunological properties, molecular biology, and genetic variability of hepatitis viruses, including hepatitis B. The detection of the Australian antigen in 1965 by Barach S. Blumberg and colleagues opened the door to many discoveries [34,35]. In 1968, Alfred M. Prince and others confirmed the presence of the “homologous serum” (HS) antigen during the acute and incubation phases [36,37], and N. Raphael Shulman and colleagues coined the term “antigen-associated hepatitis” (AAH) in 1970 [38]. The first seroepidemiological investigations of HBV were launched globally, and later, molecular biology approaches enabled the breakthrough of its cloning, sequencing [39,40], genome integration [41], and classification [42].

Prior to the deployment of HBV genotyping, as we know it today, isolates were classified into four major serotypes based on the amino acid residues at positions 122 and 160 of the HBsAg [43]. In 1988, Okamoto and colleagues sequenced the entire genome of 18 isolates, revealing an 8% intergenetic divergence distinguishing four genotypes (A-D), thus establishing the first genetic classification [44]. Norder and colleagues reported E and F genotypes in 1989 [45]. Struyer and colleagues subsequently identified genotype G in 2002 [46]. Finally, extensive studies have been conducted regarding the association of HBV with liver cirrhosis and HCC in countries with high rates of both diseases, such as Japan, China, the United States, and the European region.

## 4. Epidemiology of HBV Infections in Mexico from Serology to Molecular Biology

### 4.1. The Early Seroepidemiological Diagnostic Studies in Mexico

In parallel with the global situation, seroepidemiological research in Mexico began in the early 1970s. Table 1 summarizes some articles that provide a valuable historical reference given the risk groups and diagnostic techniques used at that time [47,48,49,50,51,52]. One study showed that professional blood donors tested at the Instituto Nacional de la Nutrition “Salvador Zubiran” in Mexico City, which cares for upper-middle and high socioeconomic class sectors, had a 2% prevalence of the HBV antigen [47].

In one of Mexico’s most extensive studies, Dr. Landa reported an average of 0.29% HBV antigen and 6.38% Anti-HBsAg in the general population [50]. However, the level of anti-HBsAg differed between high and low socioeconomic levels (7.48% vs. 3.91%), respectively.

Furthermore, other risk groups, such as healthcare workers, were highly exposed (26.4%) [51], as well as patients with clinical hepatitis [52]. These preliminary data and the future research that we see in the following section call into doubt Mexico’s classification as a low-endemicity region by the WHO.

### 4.2. Incidence and Prevalence of HBV Infection in Mexico

Estimations using data from the General Epidemiological Office (Dirección General de Epidemiologia, https://www.gob.mx/salud/acciones-y-programas/historico-boletin-epidemiologico, accessed on 5 May 2023) revealed an accumulated total of 18,846 positive cases from 1995 to 2021. Mexico City (CDMEX), Mexico State (MEX), Jalisco (JAL), Veracruz (VER), Sinaloa (SIN), Yucatan (YUC), Puebla (PUE), Baja California Norte (BC), Tamaulipas (TAM), and Sonora (SON) were the top ten states with the highest incidence, accounting for 67.09% of cases nationwide (Figure 1A, Table A1). Men were infected at a 4:1 higher rate than women, starting between the ages of 15 and 19, climbing between the ages of 20 and 24, and peaking between the ages of 25 and 44. Conversely, the women reached their highest point between 60 and 64 (Figure 1B).

At first glance, HBV infection overall appears to be a non-issue. However, dividing the data by risk groups yields a different picture, as described below (for further details, see Table A2).

#### 4.2.1. Low-Risk Groups

Beginning with the low-risk groups, HBsAg prevalence was low in the general population, at 0.46% (95% CI = 0.16–1.32) [50,53,54,55,56,57], and blood banks, at 0.21% (95% Cl = 0.15–0.29) [47,55,58,59,60,61,62,63,64,65,66,67,68,69,70,71,72,73,74,75], whereas in the former, Anti-HBc prevalence is seven times higher, at 1.43% (95% Cl = 1.05–1.95) [44,50,51,58,59]. This prevalence pattern of low antigen versus high antibodies is similar to findings reported in earlier studies [76], suggesting that HBV continues to circulate among the population but due to exposure to other risk factors, as we will mention further on.

The official picture of HBV infection is mainly built on data obtained from blood donation registries that have consistently shown a steady incidence since 1971 to date (0.21%). However, an important limitation is that only 10 out of 32 states have repeatedly published HBsAg prevalence (Figure 2), while only five states tested for Anti-HBc. Even though blood banks currently are organized as a national network, the data regarding anti-HBc and HBV-DNA are scarce or not openly accessible because these former tests, until recently, were not mandatory. This omission leads to the misleading idea that HBsAg prevalence is low. Furthermore, Sosa-Jurado and colleagues predicted a residual risk of 1 in 6410 for HBV among blood reserves, which could support the possibility of iatrogenic transmission of hepatitis B [77].

#### 4.2.2. Intermediate-to-High-Risk Group

On further literature analysis, high-risk groups (Table A2) show that HBsAg prevalence in native populations reached 7.99% (95% Cl = 5.49–11.49) [76,78,79], and in a lower proportion, hemodialysis/transfused patients at 7.24% (95% Cl = 5.66–9.21) [80,81,82,83] and psychiatric patients at 5.88% (95% Cl = 2.89–11.58) [84,85] (Figure 3). Despite the few studies on patients with liver disease, they show an HBsAg positivity rate of 5.24% (95% Cl = 3.08–8.80) [48,59,86,87,88,89] and 3.46% (95% Cl = 5.66–9.21) in patients with HIV [54,78,90,91,92,93,94,95]. However, in the latter group, studies have shown that more than half of these patients have serological evidence of past infection (anti-HBc-positive) [30].

Another way to look at these data is based on Mexico’s population’s genetic and social heterogeneity. The prevalence of HBsAg fluctuates according to socioeconomic status, and most people earn low wages. On average, HBsAg prevalence ranges from 0.46% in the upper middle class to 1.03% in the lower middle class and increases to nearly 8% in the native Mexican population. Furthermore, the proportion of anti-HBc-positive patients with prior infections is 1.43% in the upper middle class, 20.89% in the low income, and 13.88% in native subpopulations. Based on the average estimations presented in Table 2, 2.0% (2,624,570/126,699,919) to nearly 12.0% (14,764,474/126,699,919)—over 15 million people—have acquired the disease in their lifetime. Most of these infections occur among people with low income dispersed throughout the country and among natives who are mainly located in southern Mexico.

Overall, these data corroborate the observation that the prevalence of HBsAg in the general population/blood banks has remained steady [76], rising in risk groups exhibiting higher prevalence of Anti-HBc, which can approach 40% in native populations [54,79]. As a result, in the last 50 years, Mexico has had an intermediate to high prevalence of HBV infection. Because most official data published in public repositories (for example, WHO, PAHO factsheets) only represent results obtained primarily from blood banks (HBsAg), which do not reflect the real degree of HBV exposure in Mexico, it is critical to implement updated national surveys covering all risk sectors of the population.

### 4.3. Occult Hepatitis B and Misdiagnosis of HBV Infection

Occult hepatitis B infection (OBI) is a condition where HBV DNA is detectable in the blood of individuals who test negative for hepatitis B surface antigen (HBsAg) [96]. This condition is challenging to diagnose due to the absence of HBsAg and the presence of HBV DNA, and Anti-HBc may or may not be detectable. OBI can cause varying clinical impacts, including asymptomatic or severe liver diseases and a lower risk of transmission compared to active HBV infections [97]. OBI is a worldwide phenomenon described in numerous risk groups, including Mexico. As shown in Table 3, studies have been carried out in a few risk groups [67,74,75,79,91,93,95,98,99,100], in which patients with HIV and blood banks show contrasting prevalence rates.

Given the overall profile of serological markers, particularly the high rate of Anti-HBc in both the low- and high-risk groups, this may indicate that the diagnostic tests employed for screening HBsAg are not sufficiently sensitive or specific for HBV/H. Another limitation is that only HBsAg is requested in the clinical setting, without Anti-HBc or nucleic acid testing (NAT), leading to the misdiagnosis of overt and occult infection, as we have underscored previously [78].

Therefore, in the case of Mexico, the CPG should recommend testing for all three markers (HBsAg, Anti-HBc, and HBV DNA) until more sensitive and pan-genotypic diagnostic kits are manufactured. In the US, the CDC has proposed that people test for all three markers at least once or if they present risk factors [101]. These measures must be thoroughly evaluated in Mexico by researching how these markers perform in different risk groups.

### 4.4. Major Routes of Transmission and Risk Factors

As previously stated, transfusions were the predominant parenteral transmission route worldwide in the 1970s. Dr. Bernando Sepulveda indicated that Au-associated antigen hepatitis was the most severe consequence after blood transfusion. As a Mexican Academy of Medicine member, he advocated testing blood donations for Au-associated antigens and suggested removing contaminated blood units to avoid post-transfusional hepatitis [47,48]. At that time, reusable needles and syringes containing contaminated bodily fluids were a common source of infection; surgical patients and medical community members were also at high risk [52] (Table 1). Sadly, mandatory screening tests using standardized immunoassay techniques in blood banks were officially warranted until 1994 to ensure safe blood units against most blood-borne infectious diseases, including hepatitis B and C [76,102,103,104].

Once parenteral pathways were controlled, the most frequent mode of transmission has been sexual transmission, which is consistent with the pattern of incidence shown before in Figure 1. The incidence rate is high in young teens (age 21) and peaks around forty, first among men and later among women [63], a pattern that reflects the role of unsafe sex. The role of vertical mother-to-child transmission has not been frequently documented [105]. However, the high rate of HBV infections in pregnant women, HBsAg, at 0.65% (95% Cl = 0.37–1.12), and Anti-HBc, at 1.75% (95% Cl = 1.50–2.04) (Table A2, Figure 3) [55,106,107,108,109,110], may be indicative of the lack of screening and prevention measures in this group that require further investigation.

Over the last 50 years, scarce research on risk groups stands out, as illustrated in Figure 4, where 34% of the states are not represented entirely. As mentioned, blood donations were the leading cause of HBV infection, and no new risk factors for HBV emerged after 2000. However, if we use the HCV epidemiological data as a proxy, it is worth noting the increase in tattooing, piercings, acupuncture, and intravenous drug use over the last 20 years, which has caused a shift and emergence of new HCV genotypes since 1998, as shown in Figure 5 [111,112,113,114,115]. In alignment with these findings, recent studies (Table A2) have shown an HBsAg prevalence of 4.44% (95% Cl = 2.24–8.63) among drug addicts [55] and 4.82% (95% Cl = 1.82–12.15) in deferred blood donors [78]. Since both viruses share the same transmission routes, these findings may reflect the increased exposure to HBV due to parenteral drug injection. This risk behavior and the influence of migration may impact the molecular epidemiology of HBV in the future, requiring further investigation.

### 4.5. Molecular Epidemiology of HBV in Mexico

The discovery of HBV/H endemic to Mexico has not been without debate. First, in the early 2000s, Dr. Panduro’s research team reported 15 first-ever sequenced HBV isolates from hepatitis patients, of which 10 were then classified as genotype F [116]. Afterward, Arauz-Ruiz and colleagues proclaimed the new genotype H in 2002 using three samples tagged Nicaragua and Los Angeles [117]. Even in 2005, when dealing with “Hispanic” isolates from San Francisco, Dr. Misokami’s research team requested that their HBV/H samples and the new sequences be classified as F2 [118,119]. Further research demonstrated the difficulty distinguishing the so-called “divergent F strains” [120]. As more sequences were submitted, the HBV isolates formerly categorized as genotype F were recognized phylogenetically as genotype H and re-classified. Except for a few sporadic instances of this genotype in other parts of the world [121,122,123], HBV/H is the predominant genotype in Mexico since no further H isolates have been reported from Central America. Given the molecular epidemiology pattern of HBV genotypes in Mexico, it has been suggested that HBV/H is of Mesoamerican origin, as discussed further.

As depicted in Figure 6, the predominant HBV genotypes identified in Mexico uploaded to the NCBI website are H (69.8%), G (9.9%), A (7.9%), and D (7.5%), and to a lesser extent F, C, and B.

The relative distribution of these genotypes is shown in Figure 7, in which only Jalisco, Nayarit, Oaxaca, and Yucatan States have reported molecular studies. Along with the fact that some genotypes are associated with certain risk behaviors, such as G in the MSM or HIV communities [124], these data are relevant for the prognosis of the clinical course based on the population’s genetic background.

In this sense, we have hypothesized that the main clinical characteristics of HBV infections in Mexico stem from the epidemiology of HBV/H [15]; however, more research in the country’s remaining areas is required to support this idea. There is concern that a link between the HBV genotype and the population’s ancestry, as previously mentioned [78,125], could change the natural course of HBV infections. Given that the Mexican population is genetically diverse, with the north having a higher proportion of European ancestry and less indigenous ancestry, we might predict that HBV/A and D could be more common than expected. In contrast, as shown in Figure 7, HBV/H may alternate with F among the more indigenous populations of southern Mexico and Central American border countries since this transition zone served as a migration route between Mesoamerica and South America, dispersing the genotype F subtypes (F1–F4). However, further investigation is required to endorse these hypotheses.

### 4.6. Reconstruction of HBV/H Infections in the Americas

Molecular evolutionary studies on the origins and dispersion of HBV genotypes among humans and other species have been an exciting and ongoing research topic since their genotypic classification [126]. As for HBV/H, in an early study, the time to the most recent ancestor (tMRCA) for HBV/H was dated back to 1933 [127]. However, this recent introduction does not match the clinical presentation of HBV/H among the Mexican population.

Interestingly, the finding of human skeleton remains while excavating beneath Mexico City’s Ex-Hospital Real de San José de las Naturales led to the sequencing of HBV-DNA from a human tooth, revealing an HBV/H strain (SJN013) [1]. This ancestral virus sequence provided the reference to adjust the molecular clock for dating Mexican HBV sequences [128]. This remarkable anthropological and molecular discovery also broadens the hunt for the causes of the great Mesoamerican epidemics that occurred after the advent of the Spanish conquerors and whether they were responsible for the deaths of millions of natives.

The reconstruction of the HBV/H infection using Bayesian coalescence and skyline analyses suggests that the plausible route of HBV dispersion in the Americas began more than 15,000 years ago in Alaska in the form of a pre-F/H ancestor, which later separated 8000–10,000 years ago after crossing the Bering Strait and migrating southward. HBV/H likely spread from 3000 to 1500 years ago (tMRCA = 2070 years before present, 54 B.C.), coinciding with the rise of Mesoamerica’s key civilizations, including the Toltec, Olmec, Aztec, and Mayan people. It is also related to the establishment and transition to a sedentary lifestyle of these ethnic groups in Mesoamerica, which presently corresponds to the geographical region of what is now Mexico and part of Central America. Following the original dispersal, four clades (H1-H4) have emerged and may eventually evolve into subgenotypes [128].

In alignment with these findings, we have proposed that HBV/H presents various degrees of adaptation based on the current state of prevalence and clinical course of HBV infections in Mexico [129]. Incomplete virus–host adaptation results in fulminant hepatitis, whereas semi-complete adaptation results in chronic hepatitis, a more prevalent disease leading to cirrhosis and hepatocellular carcinoma. A stable state established between the HBV and the host represents a complete adaptation, which can be considered the case of OBI. Also, it is worth noting that the low prevalence of HBV-related cirrhosis and HCC in Mexico may indicate some degree of adaptation.

### 4.7. Genetic Variability of HBV/H: Antiviral Resistance and Immune Escape Mutations

HBV/H has frequently been reported to show a high variability [130]. In a recent study, HBV/H sequences showed a high frequency of antiviral resistance mutations (11.8%) and immune escape mutations (10.2%), whereas multidrug mutations accounted for 1.9% (lamivudine, telbivudine, entecavir). The percentage of immune escape mutations in other genotypes was 11.8% in HBV genotype A2, 9% in genotype F1b, and 5.6% in genotype G. Regarding antiviral resistance mutations, HBV genotype A2 had 10.22% [131]. These findings are relevant because chronic HBV infections require life-long treatments. Furthermore, immune escape mutations can influence the efficiency of diagnostic kits and vaccine immunization.

## 5. Clinical Aspects of HBV Infections in Mexico

### 5.1. HBV Infections in Cirrhosis and HCC

Cirrhosis and HCC are two of the most serious consequences of chronic hepatitis B infection. In Mexico, liver cirrhosis is the sixth leading cause of death [132]. Cirrhosis related to alcohol is the primary cause of liver disease, followed in a smaller proportion by HCV and HBV [133]. However, only a few studies indicate the etiological cause of liver cirrhosis, which seems like a paradox since the prevalence of HBV is much higher than HCV in Mexico. Moreover, although many patients are HBsAg-positive or have evidence of previous infection (Anti-HBc-positive), HBV-related cirrhosis is identified less frequently than HCV-related cirrhosis. Also, HCV patients have more severe damage when assessing liver damage with serological markers than HBV patients [134]. In other parts of the world, virus-related liver cirrhosis is associated with early contraction of HBV infection by mother-to-child transmission, mainly in highly endemic areas. Surprisingly, this transmission route remains unexplored in Mexico despite the high prevalence of HBV in pregnant women (Figure 3, Table A2). Furthermore, it may go undetected because serum HBsAg levels fall rapidly, and NAT is not routinely utilized to detect HBV-DNA, as mentioned before with OBI.

Among the studies of HBV in risk groups with cirrhosis, one earlier study found a prevalence of HBsAg of 35.2% (12/34) in alcoholic liver-diseased patients [48]. Another study indicated that 9.0% (1/11) of patients with HBV had liver cirrhosis, 45.45% (5/11) had limited localized inflammation, 18.18% had steatosis, 18.18% had chronic hepatitis, and 9% had portal hepatitis [86]. A multi-center study seeking the etiology of cirrhosis at different hospitals in Mexico found 5% (75/1486) of HBsAg-positive cases [88]. However, in another study, cirrhotic individuals had 2.82% (15/531) of HBV infection, whereas drinking (38.6%) and HCV infection (31.5%) were the leading causes of this condition [89].

The role of HBV, particularly genotype H, in the pathogenesis of HCC is also unknown in Mexican patients since no confirmatory molecular diagnostics are routinely performed. International cancer agencies report that liver cancer is more frequent than locally reported in our region, although they are often based only on clinical observations without confirming the etiological factor [135]. Furthermore, no systematic national registry is accessible to confirm these findings. One study found that 6.1% of patients with HCC were attributable to HBV, although the etiology of HCC was not detected in 30.4% of patients [136]. Outside of Mexico, only one case of HCC was reported in a 12-year-old Japanese female infected with HBV genotype H isolate presenting a PreS2 deletion mutation. However, no evidence was provided to conclude how or when the patient acquired the infection [137]. We can speculate that it does share some similarities to F1b associated with HCC in Alaska and Spain, suggesting that endemic genotypes infecting a “non-native carrier” could be more aggressive to the host.

Based on our experience and these findings, HBV infection in patients with liver cirrhosis or HCC may be underdiagnosed. Overall, there is a scarcity of publications and disparateness of HBV prevalence among the studies of both cirrhosis and HCC. Further research to clarify if HBV/H or other minor genotypes are involved and implementation of systematic measures for the early detection of HBV infection and prevention of end-stage liver disease are warranted.

### 5.2. Mixed HBV Infections

Previous research has indicated that chronic liver damage is more common in mono-infected patients with HBV genotype D than in patients infected with HBV/H [13]. However, in certain risk groups, mixed-genotype infections are common. A recent study showed that at least half of HIV co-infected patients have had HBV infection. Furthermore, among 25 HIV individuals, 44 HBV strains were identified in which HBV/H was the most common genotype, accounting for 50% of all cases, followed by G (23%), D (16%), A2 (9%), and F1b (2%). Among them, 44% were mono-infected, 36% had two HBV strains, and 20% had three HBV strains. HBV H/G genotype mixtures prevailed, followed by G/H/D, D/H, and, in lesser proportions, A2/G, A2/D/G, and H/D/F1b. Similarly, due to the mixtures among these strains, recombinants H-F (44.5%), A-G (22.5%), G-H (17.5%), and D-A (17.5%) were also found [30].

Mono-infected individuals who have HBV/H or G have lower viral loads than individuals who have a superimposed infection with two or three strains of HBV. In this group, the main risk factor associated with mixed infections was age because as they age, these patients increase their chances of new infections with other genotypes due to risk factors such as multiple sex partners and injection drug use. As the number of HBV strains rises, viral load increases and liver damage worsens, as measured by non-invasive serological indicators like FIB-4, APRI, and transitional elastography, showing diminished platelet levels [30].

Furthermore, in our experience, mono-infected patients with HBV genotypes H or G have a low or undetectable viral load and a high frequency of OBI and transaminitis, which frequently goes unrecognized. However, HBsAg positivity and persistently high viral loads may indicate liver damage and HBV mixed infection, as explained in the next section.

### 5.3. The Clinical Characteristics of HBV Infections in Mexico Do Not Match International Criteria for Treatment

First, the CPGs for the diagnosis and treatment of viral hepatitis in Mexico are typically facsimiles of international guidelines issued primarily by the AASLD, EASL, or a combination of both, and they lack evidence of the clinical and molecular characteristics of the Mexican patient and HBV/H. For example, if a patient is HBV-mono-infected, HBsAg will rapidly decline, and liver enzymes (transaminases) may not exceed 80 IU/mL. Individuals may have low levels of HBV DNA in their serum (indicating an occult B condition).

However, if viral loads rise above 1500 IU/mL, there may be undetectable clinical and laboratory liver damage unless a molecular test is conducted. If LFTs exceed 80 IU/mL and a high viral load is discovered, the patient may be in a severe stage of liver damage and have HBV mixed infection, as mentioned above. As shown in Figure 8, patients may show liver damage with a viral load lower than 1500 IU/m. International recommendations, such as those followed in the US, are a cutoff value for LFT of ≥40 IU/mL, although some treatment protocols propose 80 IU/mL and a viral load exceeding 2000 IU/m. If these criteria were applied in Mexico, many patients could continue without treatment. On the other hand, immunocompromised patients undergoing chemotherapy or organ transplantation are also at risk of HBV reactivation that may also go underdiagnosed.

Finally, as previously stated, a low detection of HCC may be attributed to the characteristics of HBV/H, which, unlike genotype F in Latin America and B and C in Asia, has not been associated with HCC. Therefore, further studies need to be performed to uncover the natural history of HBV/H to provide reliable clinical data and build national CPGs based on the Mexican patient’s genetic and environmental risk factors.

## 6. HBV Vaccine

Vaccination is the most effective defensive measure for HBV infection. According to several studies, children and young adults who follow the recommended immunization schedule (three doses) are 95% protected against contraction [138]. However, the vaccine must elicit an immunological response that produces anti-HBsAg above 10.0 mIU/mL [139].

A hepatitis B vaccine has been used in Mexico since 1999 and included in the national vaccination schedule since 2000, applied at 0 (newborns), 2, and 4 months [140]. However, the incidence data for the disease have not changed in the Mexican population despite its application for more than 20 years. As reported, only 47.7% of people aged 10 to 19 have positive anti-HBsAg (>10.0 mIU/mL and negative Anti-HBc), compared to 40.83% of people aged 20 to 25 [53]. These statistics show that fewer than 50% of young people are vaccinated against HBV. Several authors have demonstrated a problem of non-compliance with the vaccination schedule in various populations in Mexico despite regulations. A study conducted in Guerrero State in 2014 involving health workers found that only 5.5% (46/834) of the study population had completed the routine vaccination schedule against HBV [141]. Another study discussed the vaccination trend in children, finding that 71% of them between the ages of 24 and 35 months adhered to the whole vaccine schedule, compared to 65.4% of children between the ages of 12 and 23 months [142]. Furthermore, a recent study analyzing data from the National Health and Nutrition Survey 2022 reported that vaccination coverage among children from 1 to 2 years of age who completed the corresponding doses was 65.1% (95% CI 58.4–71.2) (monovalent vaccine) and 69.0% (95% CI 61.8–75.4) (pentavalent/hexavalent) [143].

As in other regions of the world, studies performed in the Mexican population with HBV/H are needed to test cross-protection of hepatitis B vaccination between genotypes that aid in developing criteria regarding the dosage and booster immunization [144,145]. Also, further studies are required to test the efficiency of the current recombinant A2-derived HBV vaccines used in Mexico, given the detection of immune escape mutations and the lack of protective levels of antibodies regardless of a complete vaccination scheme. Awareness campaigns oriented to the general public and highly exposed groups need to be reinforced to increase the coverage of this vaccine.

## 7. Discussion and Future Directions

HBsAg is the conventional marker to epidemiologically compare the prevalence rate of HBV among countries ascribed to the WHO regions. However, within the natural course of a clinical HBV infection, Anti-HBc is a useful marker because it reflects acute, chronic, and past infection, whereas HBsAg can fluctuate. In regions such as Mexico, where HBsAg is considered low, HBV infection is mistakenly not considered a health problem, and patients may go undetected with occult infection. Thus, we consider that Anti-HBc is more likely to reflect the actual degree of exposure regardless of the state of infection. However, we acknowledge that many other serum markers are emerging to diagnose and monitor HBV infections [146].

In this study, the reports included in the general population category revealed an average of 0.46% (95% Cl = 0.16–1.32) of HBsAg prevalence, whereas intermediate to high rates were seen among the risk groups. Throughout the study period, the rate of HBsAg in the general population and blood banks remained stable. However, low- and high-risk groups revealed higher rates of Anti-HBc, suggesting that the virus is still circulating among the population. Therefore, we cannot ignore the valuable information that this marker provides.

Health authorities primarily refer to HBsAg prevalence of low-risk groups as the overall infection rate in the country, masking the true extent of HBV exposure. Thus, we continue to emphasize that due to the overall lack of testing, it is more likely that the seroprevalence over the past 50 years in Mexico has fluctuated between intermediate and high rates (2.0–12.0%) of HBV infection, representing over 15 million people infected in their lifetime. During this period, HBV infection was acquired through contaminated blood donations and unsafe medical practices overlapping with sexual transmission and, most recently, by drug injection abuse. However, blood banks and healthcare professionals are currently under stricter regulations [147]. More blood donations are deferred, and contractions are avoided compared to the 1970s due to compliance with good laboratory practices and vaccines. However, a recent study has shown there is still a residual risk of HBV transmission through blood reserves [77]. 

Regarding the low HBsAg versus high Anti-HBc profile that we repeatedly find in several study groups, this can be caused by a combination of intrinsic and extrinsic factors. As we mentioned in this study, on the one hand, HBV/H is the endemic genotype whose primary host has been the native populations and eventually the admixed population of Mexico. In previous studies, we have reported that patients who resolve infection have a distinct immune response profile compared to OBI patients [148], but they are not detected due to a lack of testing. On the other hand, the current diagnostic assays for HBsAg may not be sensitive nor specific for HBV/H. In this sense, HBV/H was not considered in performance evaluation studies of HBsAg assays carried out in 2010 [149], and a marginal amount of test HBV/H samples are included when standardizing new diagnostic kits [150]. Furthermore, the WHO International Standard for Hepatitis B surface antigen (versions 1, 2, and 3) contains plasma-derived, purified, inactivated HBsAg representative of genotypes A and B [151].

In this study, vulnerable people such as pregnant women, hemodialysis patients, liver-diseased patients, HIV patients, and native communities showed intermediate to high rates of HBV infection. There are few studies or data of mothers and newborns. This fact hinders the possibility of analyzing the past attributable fraction of mother-to-child transmission (when testing and vaccines were unavailable) and the effect on potential chronic hepatitis in young children or adults. Current studies suggest that horizontal transmission is common, especially in low-income populations where children do not have complete vaccination schemes and parents are not vaccinated [105]. As proof of this point, a preliminary national serosurvey has noticed a high rate of HBsAg among samples of pregnant women (personal communication). The following three groups mentioned above are not routinely tested for HBV infection since it may depend on the head specialist treating the patient. However, we must be aware that they are multi-exposed subjects who may be at risk for mixed infections due to their state of immunosuppression. Therefore, they need better surveillance and linkage to care. In summary, Anti-HBc and NAT are not routinely requested in the clinical setting, only HBsAg, which may lead to misdiagnosis. Therefore, in the case of Mexico, the three-panel test (HBsAg, Anti-HBc, and HBV DNA) should be recommended by promoting screening campaigns and research to understand how these markers perform in different risk groups.

HBV/H is the most endemic genotype in Mexico, as shown by molecular evolutionary studies that have dated this genotype to more than 2000 years before the present. During this time, virus–host adaptations have led to HBV infections manifesting with low viral load and high rates of OBI among the native populations shown before, and apparently, HBV does not lead to chronic liver damage. However, these groups are less privileged and may not have access to vaccination or linkage to care promptly. Furthermore, other genotypes have been detected that may influence the course of infection among the admixed Mexican population. Also, antiviral resistance and immune escape mutations could provoke treatment failure and hinder the efficiency of diagnostic kits and vaccine immunization. All these conditions prompt the need to maintain molecular epidemiology surveillance.

HBV/H- or G-infected patients with chronic liver disease course present with low viral load and low LFT readings. Given these clinical characteristics, HBV infection may pass underdiagnosed if international CPGs are strictly followed. However, mixed infections should be suspected when the viral load rises, and co-infections with HCV, HEV, or HIV should be ruled out. Furthermore, Mexico has a high prevalence of HBV/G, which can cause liver damage when combined with other HBV genotypes and is primarily spread among MSM/HIV individuals. On the other hand, given the incidence data and high prevalence rates, vaccination compliance and coverage are still windows of opportunity to improve in Mexico. Thus, prioritizing diagnostics and prevention/treatment measures are necessary to avoid dissemination and undetected liver damage among risk groups.

In conjunction, these issues faced by Mexico could apply to other countries with similar economic barriers, and many of the queries and findings of this review may be transferrable. However, the nature and idiosyncrasy of each country, including Mexico, in terms of political willingness could mark the difference between lagging behind global initiatives [20,21] or opting in. In this sense, researchers play an essential role in effectively communicating the scientific data available and research needs to politicians and governments. Furthermore, the role of the local medical associations involved with viral hepatitis-infected patients can be a powerful force to set a common ground towards the development of elimination programs, which, in the case of HBV, are urgently needed.

This study exhibits both strengths and limitations that should be considered. Among its strengths, we provide valuable insights into the landscape of hepatitis B infection spanning the past five decades in Mexico. It offers insights into the country’s changing epidemiological portrait of HBV and highlights specific at-risk populations, including native groups, hemodialysis patients, individuals with liver disease, and pregnant women. Furthermore, this investigation significantly enhances our understanding of the clinical attributes associated with the prevalent HBV genotypes in Mexico, with a particular focus on HBV/H. On the other hand, the limitations of our study underscore the importance of recognizing the scarcity of publications related to HBV in Mexico and the limited availability of public databases to access morbidity and mortality due to hepatitis B. These data are key to minimizing potential errors in estimating the prevalence of HBV in Mexico, which were unavailable at the time of this study.

## 8. Conclusions

In Mexico, one of the most serious barriers to the elimination of HBV infections is diagnostics and vaccine coverage. The road to eliminating viral hepatitis is a challenge faced worldwide in which each country must evaluate the current state of strengths and weaknesses when elaborating the programs that potentially will fulfill that goal. One first step is having a clear picture of the degree of information and knowledge regarding the epidemiological, molecular, and clinical characteristics of HBV/H infections to prioritize actions such as training, research, and awareness. These actions should create more specific CPGs according to the region’s characteristics. Mexico’s elimination plan for HBV will require teamwork among government health administration, researchers, physicians, specialists, and civil society advocates to overcome this task jointly.

## Figures and Tables

**Figure 1 viruses-15-02186-f001:**
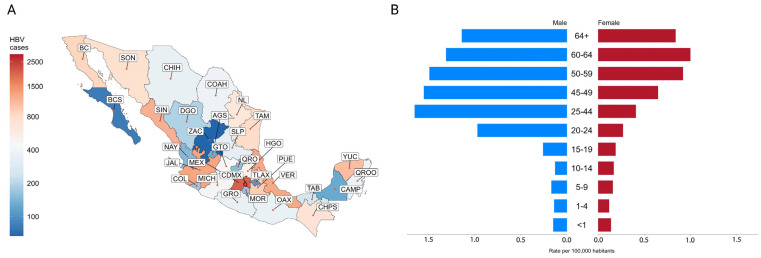
Incidence of HBV infection cases in Mexico. (**A**) Heatmap of the accumulated HBV incidence reported by state from 1995 to 2021 (reference: National System of Epidemiological Surveillance, SINAVE, Table A2). (**B**) Accumulated incidence rate of HBV cases per 100,000 inhabitants stratified by gender and age from 2000 to 2021 (reference: National System of Epidemiological Surveillance).

**Figure 2 viruses-15-02186-f002:**
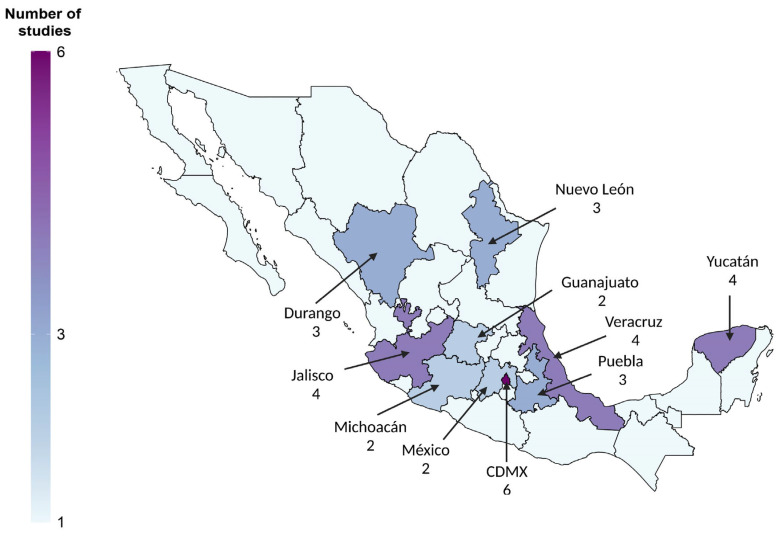
Heatmap of the number of studies reported by blood bank entities nationwide from 1971 to 2022 regarding HBsAg prevalence.

**Figure 3 viruses-15-02186-f003:**
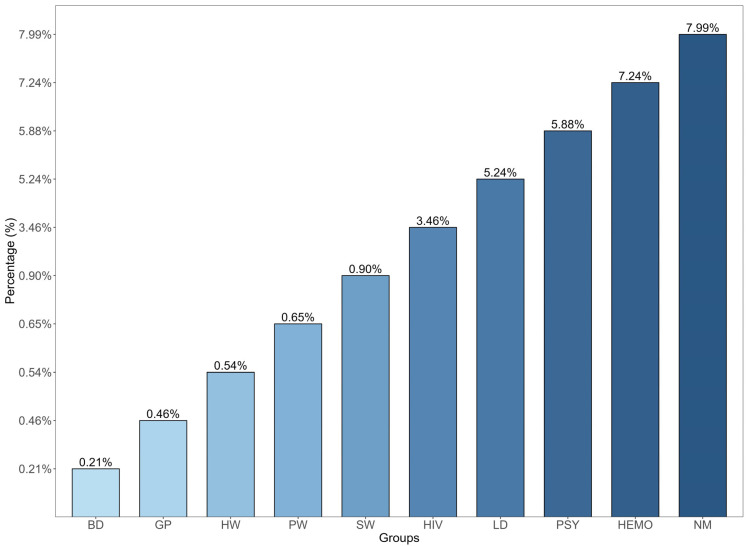
Prevalence of HBsAg in representative low- and high-risk groups. BD, blood donors; GP, general population; HW, healthcare workers; PW, pregnant woman; SW, sex workers; HIV, human immunodeficiency virus patients; LD, liver disease; PSY, psychiatric patients; HEMO, hemodialysis/transfused; NM, native Mexican. Source: Table A2.

**Figure 4 viruses-15-02186-f004:**
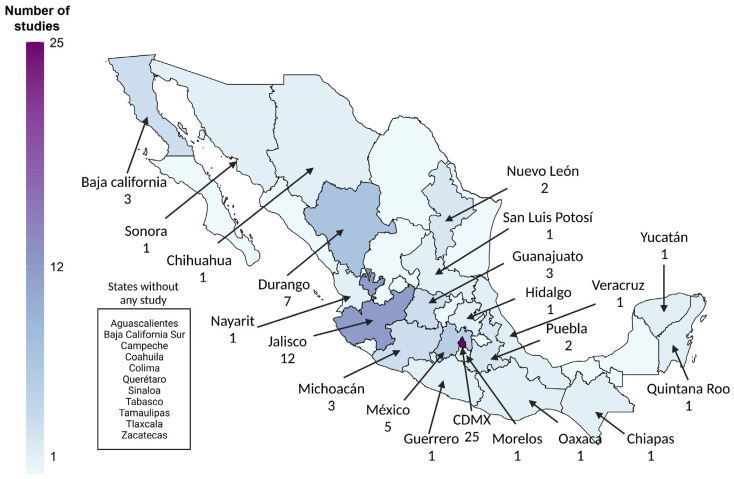
Heatmap of the number and distribution of studies reporting risk factors for HBV infection nationwide from 1971 to 2022.

**Figure 5 viruses-15-02186-f005:**
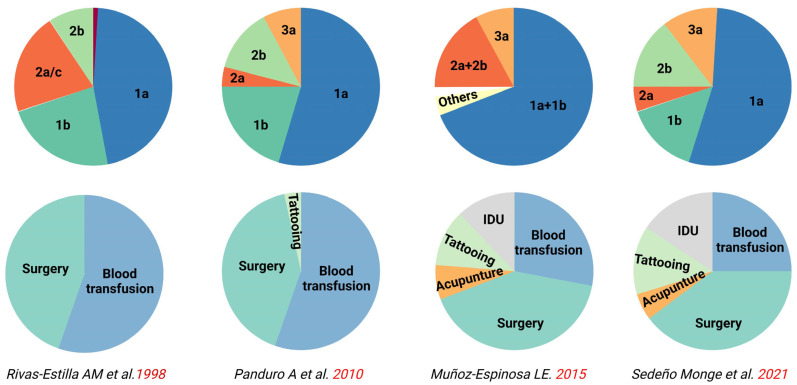
Timeline showing the shift of HCV risk factors associated with HCV molecular epidemiology from 1998 to 2021 that may influence the risk for HBV infections. (The upper section shows pie charts of the distribution of HCV genotypes versus reported risk factors in the lower section. Source: [111,112,113,114,115]).

**Figure 6 viruses-15-02186-f006:**
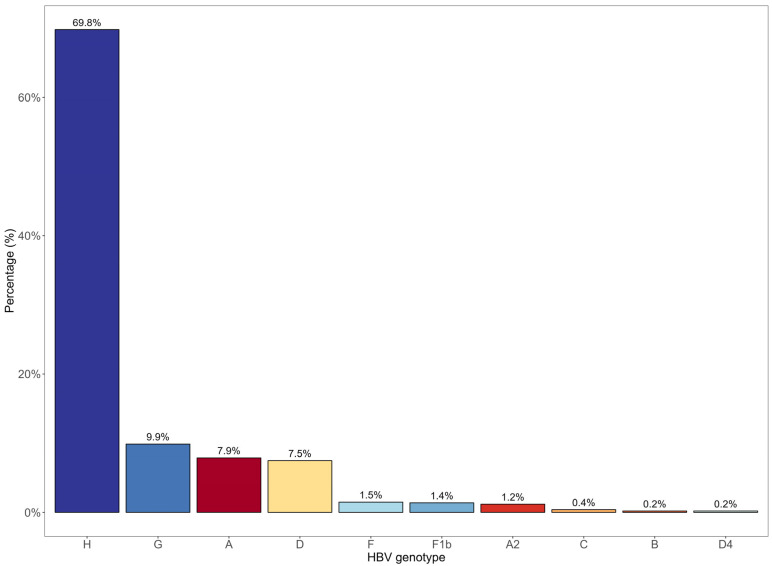
Distribution of the HBV genotypes isolated in Mexico. Resource: NCBI (Genbank, 2022).

**Figure 7 viruses-15-02186-f007:**
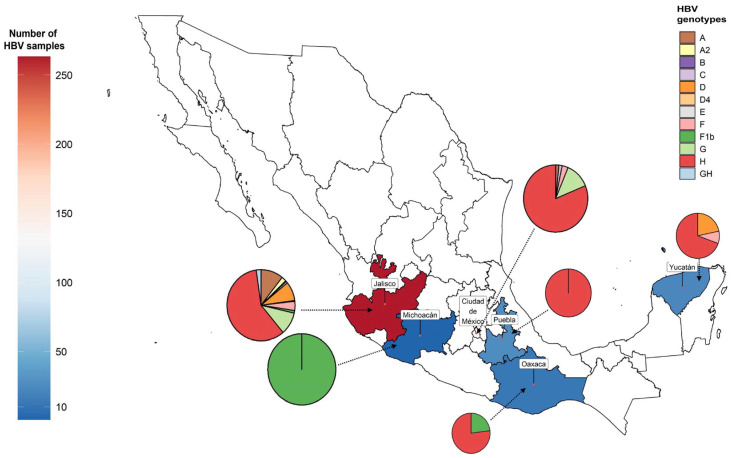
Heatmap showing the number of studies and the relative geographic distribution of HBV genotypes isolated in Mexico. Resource: NCBI (Genbank, 2022).

**Figure 8 viruses-15-02186-f008:**
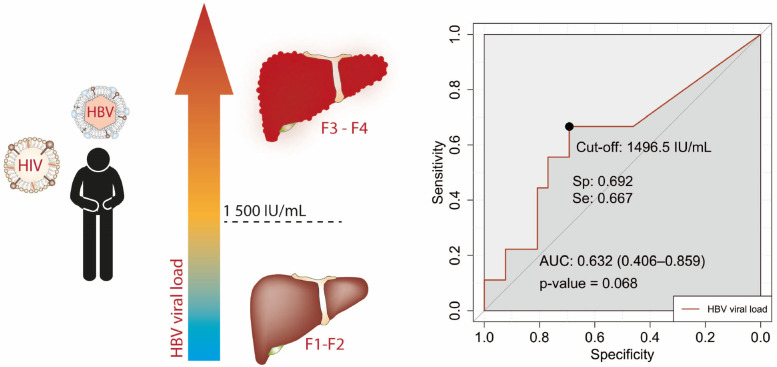
HBV viral load related to advanced liver damage among patients with HIV. In a cohort of chronically infected hepatitis B patients co-infected with HIV, liver stiffness and HBV viral load levels were assessed (*n* = 35). This analysis revealed a correlation between an HBV viral load of approximately 1500 IU/mL and advanced liver damage (F2–F3). (See Section 2 and reference [30] for more details).

**Table 1 viruses-15-02186-t001:** Early HBV seroepidemiological studies carried out in Mexico in different risk groups.

Study Population	Subjects (N)	Antigen + n (%)	Antibody + n (%)	Diagnostic Technique	Year/Ref.
Professional blood donors	4196	28 (0.66) (AAH)	7 (0.16)	IEP	1971/[47]
Professional blood donors Voluntary blood donors	551 594	11 (2.0) 0 (0.0)	- -	CF	1971/[47,48]
Children with high values of liver function tests Children with lymphomas/ Leukemias	- -	5.7% 11.7%	- -	ID	1972/[49]
General population Netzahualcoyotl City, Mexico State San Angel, Mexico City	19,249	0.29% - -	6.38% 7.48% 3.91%	IEP	1976/[50]
Healthcare workers Non-healthcare workers	545 500	143 (26.4) 72 (14.4)	-	RIA	1984/[51]
Patients with acute hepatitis	222	11(5.0)	-	RIA	1987/[52]

Abbreviations: Ref., reference; N/n, number; AAH, antigen-associated hepatitis; IEP, (high-voltage) immunoelectrophoresis; CF, complement fixation; ID, immunodiffusion; RIA, radioimmunoassay.

**Table 2 viruses-15-02186-t002:** Estimations of the number of potential infections based on HBsAg/Anti-HBc prevalence and income status.

Subpopulations *	Inhabitants (N)	HBsAg Prevalence ** (%)	Inhabitants (N)	Anti-HBc Prevalence ** (%)	Inhabitants (N)
Middle/High Income	51,782,209	0.46	238,198	1.43	740,485
Low income	51,717,710	1.03	532,692	20.89	10,803,829
Native Mexicans	23,200,000	7.99	1,853,680	13.88	3,220,160
Total	126,699,919		2,624,570		14,764,474

* Data according to INEGI; ** average; N, number; source: Table A2.

**Table 3 viruses-15-02186-t003:** Prevalence of OBI among low- and high-risk groups.

Study Groups	Studies (N)	Prevalence OBI % (95% CI)	Prevalence Anti-HBc % (95% Cl)	Reference
HIV/AIDS	3	34.30 (19.76–52.52)	22.26 (15.67–30.61)	[91,93,95]
Native population	1	14.19 (10.62–18.70)	32.67 (27.66–38.13)	[79]
Children	1	11.16 (2.30–22.0)	9.5 (6.08–13.98)	[99]
Transplanted patients	1	2.74 (0.69–10.30)	0 (0.04–9.89)	[100]
Blood Donors	4	2.50 (2.30–22.0)	1.45 (1.10–1.91)	[67,74,75,98]

Abbreviations: OBI, occult hepatitis B; N, number; CI, confidence interval.

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
