# Peer review of "Hepatitis B Virus Genotype H: Epidemiological, Molecular, and Clinical Characteristics in Mexico"

_viruses, 2023, doi:10.3390/v15112186_

Round 1

Reviewer 1 Report

Comments and Suggestions for Authors

General Comments

  Panduro and colleagues have compiled a review article surrounding the molecular epidemiology and clinical characteristics of HBV GtH. They have summarised many helath reports from previous studies as well as heath care providers/centres to asses the disease burden and the social factors with which it correlates in Mexico. At a glance, this may seem somewhat niche, however given the well defined geographical distribution of HBV Gts, this study is well placed to consider the data.

Panduro et al have conducted their study thoroughly, and overall the manuscript is very well written, and exhibits high quality written English throughout. The review is very well crafted and does well to explain the possible biases surrounding the studies which they have interrogated.

It is a pity that the figures have been compiled into the PDF at the expense of the resolution, and the image quality is not quite as good as one would hope, but this is an issue to be resolved by the publication team. It is a shame as the plots seem are very well compiled, they are just largely unreadable. However, the figure legends are often lacking detail. The authors should provide a more comprehensive description of the figures to aid the reader.

In short, thank you for putting together such a comprehensive review article. The work is well written, and will prove to be a valuable resource for the field.

1.     Introduction

Whilst I understand this is largely a clinical review, it would seem like an omission to not briefly touch upon the molecular basis for calling of genotypes and sub-genotypes in terms of genetic diversity in the introduction. Given the extensive body of work contributed, it is surprising to see that there are no citations from Kramvis in this review. The authors could also comment on the debate surrounding the existence of Gt-I and -J, where there is still some controversy surrounding their classification. When citing the work surrounfing paleogenomics, it would be worth paying credit to the study from Muhlemann, who investigated bronze age isolates.

The paragraph starting from line 50 refers to some of the hurdles. These were explored in a review article by Peter Revill, where the main obstacles to cure were outlined. This work should be cited, and probably described here.

The authors could probably acknowledge that the issues faced in Mexico would be largely applicable to other countries with similar economic barriers, and many findings of this review would be transferrable.  

2.     An overview of the major milestones of viral hepatitis

Are Hippocrates’ observations exclusive to viral hepatitis? Surely jaundice could be reflective of several other liver diseases? Rephrase the final sentence, line 99- as Europe is a not a country, where the other listed items are.

3.     Epidemiology of HBV infections in Mexico from serology to molecular biology

3.1: It is interesting that the data presented in table 1 are reflective of patients in the middle classes. Was there any speculation in these studies as to the disease burden amongst those who were less financially privileged? The difference between high and low socioeconomic levels appears to be taken from the difference between samples from San Angel and Netzahualcoyotl City – for those of us who are unaware of the Mexican wealth distribution, further clarification may be needed. Furthermore, the healthcare statistic is not surprising, but still a hugely overrepresentation of HBs+: is it possible to determine whether the geographical distribution of those workers is also split between those tending to upper and lower incomes?

3.2: Is the classification of HBV prevalence based on HBsAg or Ab? It may be a mix, or from medical records but it would make sense to state the method by which HBV infection was called. Figure 2: What does the colour scale represent? There is no title to the scale bar, and the legend also lacks detail on that. This is a very interesting finding – the mandatory reporting of HBs Ag/Ab surely is worth noting from these institutions!

Figure 3: Given how previously healthcare workers represented c25% of infections, is it surprising that they seem to show a reasonably low infection rate in table A2?

Line 174: authors need to revise the formatting of numbers and be careful with the transposition of ' and , when listing 1,000,000 etc. This issue also persists in Table 2, and may occur in other places in the review, please check.

Could the authors comment further on the discrepancies between HBsAg prevalence and anti-HBc? Surely, if the field is considering HBsAg clearance as indicator of a functional cure, we may be underestimating the burden  of disease? This imbalance between HBs and anti-HBc would be worth considering worldwide, and not limited to just Mexico?

Section 3.3 is extremely put together and gives a succinct and considered summary, well done.

      It is interesting that the vertical transmission route has not been frequently documented in these studies. Generally, it is considered that this would be a major route of spread, and pose a huge obstacle to a functional cure. Do the authors think that this may be a genotype specific phenomenon? Or are the reports in Mexico on Gt-H just fewer than the other genotypes in other regions? It may be worth considering moving the text at line 237 up to before 227, it currently reads as though HCV is a typographically error, and the section at 237 provides more context to the review.

Figure 4: It is a pity resolution is lost during compression or export, but hopefully this can be resolved. Title or label for scale bar is missing once again.

Figure 5: Authors should add more detail to the legend to explain what the top and bottom pie charts are showing. One assumes its genotype, but clarity would be appreciated.  

3.5: the definition of genotype classification is listed here, but it would still be better suited to the introduction section. ‘Norder H’ and colleagues (l252) is a little difficult, given it falls in a section about GtH! Authors could change to ‘H Norder’ or simply ‘Norder’ to improve readability.

Figure 6: Is there any reason why the authors have presented some sub-genotypes rather than pooling into genotypes? For example A and A2, and F and F1b could be merged. Minor edit, and by no means essential.

Figure 7: It is a pity that the resolution of this figure has severly impacted the interpretability of these data. It is not clear what the ‘HBV genotyped’ actually refers to. Perhaps percentage values of the pie charts could also be annotated. This figure would benefit from a more comprehensive legend.

The final paragraph of 3.5 is very well compiled, if a little light on the number of primary articles cited. Is there any evidence of co-infection with multiple genotypes? The authors have provided an excellent write up on the reconstruction of HBV/H infections in 3.6; with no revisions required.

4.     Clinical aspects of HBV infections in Mexico

The authors should be commended on their work in this section; the review is thorough and provides a detailed summary of the link between GtH infection and HCC, as well as cirrhosis and co-infection with HCV.

            The text in section 4.2 renders my previous questions on multiple genotype co-infection obsolete. Is it the authors belief that increased sexual encounters are the main reason for superimposed infection? Is it also possible that increased drug usage, increased medical attention, dialysis and transfusions could also be implicated?

            Figure 8 and the associated text are of great interest, and there is a growing interest in the field in HIV/HBV co-infected patients– largely due to the ART patients receive. Do most patients in Mexico present with HIV symptoms and get diagnosed with HBV during tests? There is of course some overlap in activity of RT-inhibitors for HBV and HIV, and there may be differences in rug efficacy across diverse HBV genotypes – however exploring this is probably beyond the remit of this review.  

5.     HBV Vaccine

Have there been any studies to investigate the activity of Anti-HBsAg antibodies generated against a panel of diverse HBV genotypes. The authors should comment on how the HBV vaccine is generated, and how genotype agnostic that strategy is.

Can the authors provide any statistics to show the levels of vaccine uptake within the population? Is there a high level of vaccine hesitancy in the lower economically advantaged population of Mexico?

What is the Mexican policy for compulsory vaccination of health care staff and public servants, does that explain the decline in infection in healthcare workers presented in section 3?

6.     Conclusion and Future Directions

Overall the conclusions are well considered, and accurately represent the data presented and findings discussed in the rest of the manuscript. In the introduction, the authors discussed obstacles to cure, and these have been previously reviewed by Peter Revill. It would be interesting for the authors to compare their suggestions for a change in Mexican policy and strategy to the more general suggestions made by Revill et al. Are their geographical and genotype specific recommendations, or is global cure hampered by the same hurdles?

Reviewer 2 Report

Comments and Suggestions for Authors

The paper by Bernasconi E. et al. is an integrative literature review describing strengths and weaknesses in diagnostics and prevention measures that explain the current epidemiological profile of hepatitis B virus genotype H (HBV/H) in Mexico. The authors concluded that training, research, and awareness actions are required to control HBV infection in Mexico.

The review is well-done but some changes are necessary to be accepted in viruses

Comments:

·       General comment. As HBsAg prevalence in the general population is 0,46% (95% CI=0.16-1.32)[34,37–41] the authors should not conclude that Mexico has an intermediate to high prevalence. Authors could say that Mexico has groups with high risk but the prevalence in the general population is low.

·       Title. Please, the title should include “….in Mexico” for a better description. 

·       Abstract. Please, include the methods to review the data (see “methods section”). 

·       Methods and data analysis. Please, include a methods section describing the methodology used to do this review (doi: 10.1111/scs.12327)

·       Introduction. Information regarding the HCV virus is not the subject of this review. Authors could eliminate the lines 60-71

·       Findings. Epidemiology. Please separate the studies regarding the general population and the studies about risk groups in different paragraphs and change the title and lines 178-182 about the “high prevalence of HBV” in Mexico. Clinical aspects. Please, avoid recommendations without results or references (lines 363-364 and 386-390 and lines 411 and 415)

·       Tables

·       Figures 2. For a better understanding separate into 3 groups (low, general, and high)

·       Discussion and Future directions. Please change line 444 according to the HBsAg prevalence in the general population. Please include the strengths and limitations of this review.

·       Conclusion. Please separate in a new paragraph the conclusion of this review

Round 2

Reviewer 2 Report

Comments and Suggestions for Authors

The authors have made modifications to the original manuscript based on the reviewers’ comments and advice improving the quality of their study. 

Now, the manuscript is suitable for publication in viruses